# Syncytiotrophoblast Markers Are Downregulated in Placentas from Idiopathic Stillbirths

**DOI:** 10.3390/ijms25105180

**Published:** 2024-05-09

**Authors:** Sara Vasconcelos, Ioannis Moustakas, Miguel R. Branco, Susana Guimarães, Carla Caniçais, Talia van der Helm, Carla Ramalho, Cristina Joana Marques, Susana M. Chuva de Sousa Lopes, Sofia Dória

**Affiliations:** 1Genetics Service, Department of Pathology, Faculty of Medicine, University of Porto, 4200-319 Porto, Portugalcmarques@med.up.pt (C.J.M.); 2i3S—Instituto de Investigação e Inovação em Saúde, University of Porto, 4200-135 Porto, Portugal; 3Department of Anatomy and Embryology, Leiden University Medical Center, 2333 ZC Leiden, The Netherlandst.van_der_helm@lumc.nl (T.v.d.H.); s.m.chuva_de_sousa_lopes@lumc.nl (S.M.C.d.S.L.); 4Sequencing Analysis Support Core, Leiden University Medical Center, 2333 ZC Leiden, The Netherlands; 5Blizard Institute, Faculty of Medicine and Dentistry, Queen Mary University of London, London E1 2AT, UK; 6Department of Pathology, Faculty of Medicine and Centro Hospitalar Universitário São João, 4200-319 Porto, Portugal; 7Department of Obstetrics and Gynaecology, Faculty of Medicine and Centro Hospitalar Universitário São João, 4200-319 Porto, Portugal

**Keywords:** idiopathic stillbirth, placenta, transcriptome, syncytiotrophoblast, cytotrophoblast

## Abstract

The trophoblast cells are responsible for the transfer of nutrients between the mother and the foetus and play a major role in placental endocrine function by producing and releasing large amounts of hormones and growth factors. Syncytiotrophoblast cells (STB), formed by the fusion of mononuclear cytotrophoblasts (CTB), constitute the interface between the foetus and the mother and are essential for all of these functions. We performed transcriptome analysis of human placental samples from two control groups—live births (LB), and stillbirths (SB) with a clinically recognised cause—and from our study group, idiopathic stillbirths (iSB). We identified 1172 DEGs in iSB, when comparing with the LB group; however, when we compared iSB with the SB group, only 15 and 12 genes were down- and upregulated in iSB, respectively. An assessment of these DEGs identified 15 commonly downregulated genes in iSB. Among these, several syncytiotrophoblast markers, like genes from the *PSG* and *CSH* families, as well as *ALPP*, *KISS1*, and *CRH*, were significantly downregulated in placental samples from iSB. The transcriptome analysis revealed underlying differences at a molecular level involving the syncytiotrophoblast. This suggests that defects in the syncytial layer may underlie unexplained stillbirths, therefore offering insights to improve clinical obstetrics practice.

## 1. Introduction

Stillbirth is a significantly common adverse outcome of pregnancy, inflicting a stressful and traumatic experience upon parents [1]. It occurs in up to 3.3 per 1000 births in Europe [2,3], affecting over 7000 women worldwide every day [4]. The most prevalent risk factors associated with stillbirth include advanced maternal age, obesity, pre-existing diabetes, chronic hypertension, smoking, alcohol use, having a pregnancy using medically assisted reproduction, multiple gestation, male foetal sex, and past obstetrics complications [1]. According to the International Classification of Diseases, stillbirth is defined as the demise of a foetus weighing more than 500 g or surpassing 22 weeks of gestation [4,5]. Causal pathways for stillbirth frequently involve impaired placental function, either with foetal growth restriction (FGR), preterm labour, or both [6].

The placenta is a complex and transient organ crucial for foetal growth and development, being essential for a successful pregnancy. Placental development involves processes such as trophoblast invasion, proliferation, and differentiation, as well as vasculogenesis and angiogenesis, ensuring adequate blood supply to support foetal growth [7]. The placenta regulates the exchange of nutrients, gases, and waste products between the maternal and foetal blood, shielding the foetus from potential harm and maternal immune reactivity, and synthesises and secretes hormones essential for maintaining the pregnancy. The syncytiotrophoblast (STB) layer is the primary placental cell layer responsible for these functions [8]. During pregnancy, the STB undergoes continuous growth through the differentiation and fusion of cells in the underlying progenitor cell layer of cytotrophoblasts (CTBs). Together, the STB and the CTB, both trophectoderm-derived cells, form the outer part of the villous structure. The STB communicates with maternal blood in the intervillous space, while the villous core is formed by extraembryonic mesenchymal cells and the foetal vasculature. As pregnancy progresses, syncytial knots form through the accumulation and budding off of syncytial nuclei in the STB [9,10], appearing transcriptionally inactive and exhibiting evidence of oxidative damage [11]. Dysregulated STB formation can compromise the integrity of the placental exchange surface and negatively impact the foetal development [12]. Moreover, numerous factors produced by the STB enter into the maternal circulation, significantly influencing the maternal physiology, contributing to maintaining a healthy pregnancy. These factors encompass steroid and peptide hormones (e.g., oestrogen, progesterone, human chorionic gonadotropin (hCG), human placental lactogen (hPL), and placental growth hormone (PGH), as well as pregnancy-specific glycoproteins (PSGs), vascular endothelial growth factor (VEGF), placental growth factor (PlGF), and TGFβs [7,8]. Given the intricate functions and factors released by the placenta, disruptions in placental development have been linked to several pregnancy complications and compromised foetal outcomes [7].

The impact of placental abnormalities on the aetiology of stillbirth is indisputable, as supported by observations of placental pathology in several clinical conditions linked to increased stillbirth rates, including FGR, pre-eclampsia, and placental abruption [13]. Interestingly, stillbirth demonstrates a strong association with FGR, exhibiting a considerable overlap in risk factors and potential causes between stillbirth and FGR [14,15]. Identified causes of stillbirth encompass conditions affecting the mother and/or the foetus, such as maternal hypertension, foetal–maternal haemorrhage, congenital anomalies, infections, and umbilical cord abnormalities [14,16].

In addition to cases with identified aetiopathology, a significant proportion of stillbirths remain unexplained, with some presenting placental histological alterations of uncertain clinical significance [17]. Because of the large number of classification systems that have been developed based on different clinical approaches, definitions, and levels of complexity, the proportion of stillbirths of unknown aetiology is difficult to estimate [18], ranging from 15% to 71% in the literature [5,19,20,21]. A comprehensive review spanning 85 reports across 50 countries revealed that “unexplained” was the most commonly reported cause of death in more than 400,000 described stillbirths [20]. For these unexplained cases, various factors, including placental insufficiency, may contribute to an intrauterine environment that is detrimental to the foetus [22]. Some studies have shown that these cases are often associated with FGR [23,24].

Throughout pregnancy, the placenta undergoes continuous changes in gene expression pathways, facilitating the regulation of foetal growth, establishment of immunological tolerance, and modulation of metabolism in alignment with the demands of the pregnancy [25]. Therefore, analysing the dynamics of the placental transcriptome can offer insights into the dynamics of placental function throughout pregnancy [1,26]. While a substantial amount of placental “omics” studies have focused on idiopathic pregnancy losses in the first trimester [27], there is a notable gap in comprehensive analysis of gene expression during the third trimester. By exploring the transcriptomes of placentas from idiopathic stillbirths (iSB), we identified placental pathways and molecular mechanisms underlying idiopathic causes. This pursuit holds the potential to unveil critical markers for helping elucidate idiopathic cases. Indeed, understanding the aetiology of stillbirth is essential for providing better counselling to affected parents, preventing recurrence [28], and enhancing strategies to optimise future pregnancy outcomes.

## 2. Results

### 2.1. Placental Samples Have Heterogeneous Gene Expression Patterns

Gestational ages ranged between 26 and 39 weeks of gestation, with no statistically significant differences in gestational age or maternal age between the groups (Appendix A). We performed bulk RNA-Seq of human placental samples—three live births (LB), five stillbirths (SB), and four idiopathic stillbirths (iSB)—to identify gene expression changes associated with iSB. The heatmap for the top 50 most highly variable genes across all placental samples revealed two major clusters—Cluster 1, comprising two SB and two LB placental samples (Figure 1A, left); and Cluster 2, with samples from the three biological groups—one LB, three SB, and four iSB (Figure 1A, right). Principal component analysis (PCA) further showed considerable heterogeneity among the SB and LB placental samples. Nonetheless, the iSB cases were closely positioned, with three of the four samples falling within the same quadrant (Figure 1B). Thus, although our control placentas, SB and LB, showed heterogeneity, our idiopathic cases were very similar to each other.

### 2.2. Differentially Expressed Genes in Idiopathic Stillbirth

Next, we analysed differential gene expression between the groups (Figure 2 and Appendix A).

Initial comparison between placental samples from SB and LB (representing the control groups) revealed a total of 414 differentially expressed genes (DEGs) (FDR < 0.05), with 238 downregulated in LB and 176 upregulated in LB (Figure 2A). As expected, GO enrichment analysis of the obtained down-DEGs (such as *CD180*, *IL6ST*, and C-X-C or C-C motif chemokine ligands—*CXCL3*, *CXCL8*, *CCL3L1*, *CCL4L2*, *CCL4*) unveiled enrichment of “inflammatory response” and “immune system process” in SB, reflecting the aetiology of SB samples, with foetal loss due to acquired infections. In fact, the affected KEGG pathways were related to the Toll-like receptor signaling pathway, B-cell receptor signaling pathway, and viral protein interactions with cytokine and cytokine receptors. Conversely, LB exhibited enrichment in biological processes associated with “cellular signaling" (*LEMD2*, *MAP3K11*, *SORBS3*, *PPP1R12C*, *ADAM15*, *JAG1*) (Figure 2B).

Secondly, comparison between placental samples from iSB and LB revealed 1172 DEGs, with 660 genes downregulated and 512 genes upregulated in iSB (Figure 2C). GO analysis indicated that downregulated DEGs in iSB (such as *KISS1*, chorionic somatomammotropin hormones (*CSH1* and *CSH2*), *SMAD6*, and *SYDE1*) were associated with cellular developmental processes, such as “cell differentiation” and “regulation of signaling” (Figure 2D). Notably, the only KEGG pathway affected here was focal adhesion. Upregulated DEGs in iSB (such as the hypoxia-inducible factor *HIF1A*, *NRF2*, *CYP1A*, and *SOD2*) were also related to regulation of developmental processes, with specific terms for “response to organic substance” and “response to endogenous stimulus” (Figure 2D). Finally, the comparison between iSB and SB revealed 27 DEGs, of which 15 were downregulated in iSB and 12 were upregulated (Figure 2E). Amongst the most downregulated genes, several pregnancy-specific glycoproteins (PSG1, *PSG9*, *PSG8*, *PSG4*, *PSG6*, and *PSG3*), different hormones (*GH2* (growth hormone 2), *CRH* (corticotropin-releasing hormone), *CSHL1*, and *CSH2*), and other placental markers (*ALPP* (placental alkaline phosphatase) and *KISS1*) were observed, which could be associated with the downregulated reproductive biological processes observed, like “response to growth hormone” (Figure 2F). In fact, in the previous comparison, *PSGs*, hormones, and placental markers were also found to be downregulated in iSB. *PSG* genes were further studied in a greater number of samples (including transcriptome samples) by real-time qPCR (Appendix A). Interestingly, the affected KEGG pathway associated with upregulated genes was related to the regulation of heat-shock response and cellular response to heat stress. Other enriched GO terms can be found in Appendix A.

### 2.3. Idiopathic Stillbirth Placental Samples Showed Dysregulation of Syncytiotrophoblast Markers

Importantly, dysregulated genes identified in the comparison between SB and iSB were not present in the list of DEGs between SB and LB, whereas 24 genes were also present in the iSB vs. LB comparison. Therefore, focusing on placentas from idiopathic cases, we identified 24 dysregulated DEGs that were common between the two comparisons (iSB vs. LB and iSB vs. SB) (Figure 3A). Amongst these, and as stated above, several members of the *PSG* and *CSH* families, as well as *ALPP*, *KISS1*, and *CRH*, were significantly downregulated in placental samples from iSB. As these genes are highly expressed in the STB, we hypothesised that this was indicative of cell composition differences between samples, perhaps due to defects in the syncytial layer in iSB placentas.

To test this hypothesis, we first compared the expression of all 24 DEGs between CTB and STB using published RNA-Seq data [29]. This showed that iSB down-DEGs were specifically expressed in STB, whereas iSB up-DEGs were preferentially expressed in CTB (Figure 3B). We then leveraged single-cell RNA-Seq data to expand our analysis to all cell types present in term placentas [30]. We extracted the marker genes of each cell type and asked if their expression was changing in iSB when compared to LB (Figure 3C) or SB (Figure 3D). The results show that whilst the marker genes of most cell types were either stable or had slightly increased expression in iSB, those associated with STB were the only group of marker genes that displayed decreased expression in iSB (adjusted *p*-values for all cell types can be found in Appendix A), with the effect being more pronounced when compared against LB samples. Plotting the expression of each STB marker gene in iSB relative to the control groups confirmed that most STB markers were downregulated in our idiopathic cases, corroborating the observed results (Figure 3E,F). Altogether, these findings suggest a dysregulation of the STB layer in iSB placentas.

### 2.4. Immunohistochemical and Immunofluorescence Analysis

To assess potential disparities in the STB layer in our idiopathic cases, we analysed five placental samples: two iSB (one of them with FGR), two SB (foetal deaths caused by an infection and by an umbilical cord pathology), and one LB (Figure 4). Ten randomly selected fields of view were assessed for the images of each placental sample. As all ten fields were highly similar, we selected one representative of the group, which was also chosen for the immunohistochemistry and immunofluorescence slides.

In placentas from SB and from LB, hCG staining was normally found on the membranes of the STB layer of the villous surface. In the iSB sample also presenting FGR, we observed an apparent increase in syncytial knots with a small and dark aspect on the surface of the chorionic villi, which were intensely marked by the hCG. Regarding the other iSB case, we observed focal areas without hCG staining interspersed with areas that stained weakly. The enrichment of the term “cell population proliferation” in iSB also led us to evaluate the proliferative marker Ki67 staining, which marked proliferative CTB. Indeed, a residual number of CTBs are expected in a term placenta, as observed in LB. Interestingly, iSB1 resembled LB and, therefore, did not exhibit changes in proliferative CTB. However, we observed more positive CTB in iSB with FGR compared to LB, but an absolute quantification would be necessary to confirm this increase. Considering all of the results, we postulated that there is less viable STB in iSB; however, it affected different pathways, indicating a cell composition change in iSB and a more complex aetiology. A larger sample size and an absolute quantification will be crucial to better understand how the STB is affected in iSB samples.

## 3. Discussion

Stillbirth is a common occurrence but is one of the least studied adverse pregnancy outcomes [2], compared to other pathologies such as pre-eclampsia, preterm labour, and recurrent miscarriage [31]. The main goal of our study was to identify genes with altered expression in placentas from iSB and thereby unravel the molecular mechanisms underlying this clinical condition.

Vasilis Sitras et al. observed that more than half of the genes expressed in placenta (7519 out of 12000) undergo changes in expression from the first to the third trimester of pregnancy, confirming that the placenta undergoes a profound molecular rearrangement in order to adapt to the changing demands of the foetus [32]. The most substantial differences in the transcriptomes of idiopathic cases were observed when compared with placentas from healthy term live births. GO analysis of iSB vs. LB revealed a downregulation of cell differentiation in iSB, suggesting potential defects in the STB layer. In the comparison between SB and iSB, several GO terms related to the response to growth hormones were significantly downregulated in iSB placentas. Since STB plays a key role in endocrine function during gestation, these findings also suggested a potential deficiency in STB function. Notably, several STB markers were downregulated in idiopathic cases compared with both LB and SB. Therefore, it appears that there is a problem with the STB that is unrelated to whether it is associated with pregnancy loss or a live birth, and it does not seem to be linked to potential differences in gestational age, since these STB markers were not present in the comparison between SB and LB.

An illustrative example of downregulated genes in unexplained stillbirths is *ALPP*, which is a reported STB marker in term pregnancies [33,34]. Bellazi and colleagues suggested that *ALPP* could also be a positive regulator of placental growth and may act as proliferative factor in human trophoblastic cells, influencing foetal growth and development [35]. The dysregulation of *ALPP* expression in pregnancy losses is still debatable, but some *ALPP* polymorphisms have been associated with spontaneous abortions [36], and abnormal ALPP levels have also been implicated in FGR [37]. Another gene, *KISS1*, which is abundantly expressed in the STB [38], was also observed to have significantly lower expression levels in patients with first-trimester pregnancy losses, compared to those requesting elective terminations [39,40]. Furthermore, the human growth hormone/chorionic somatomammotropin hormone (*hGH*/*CSH*) locus at 17q22-24, consisting of one pituitary-expressed postnatal gene (*GH1*) and four placenta-expressed genes (*GH2*, *CSH1*, *CSH2*, and *CSHL1*), is implicated in the regulation of postnatal and intrauterine growth [41,42,43]. Interestingly, all of these genes of the placental *hGH/CSH* cluster also exhibited a dramatic decrease in the idiopathic cases analysed. The expression of the placental *hGH/CSH* cluster was studied in conditions such as pre-eclampsia and gestational diabetes mellitus, showing a trend for reduced transcript levels in these pathological conditions [42]. Additionally, Liu et al. observed a decrease in CSH1 protein levels in the villi from early pregnancy loss [44]; however, as far as we are aware, there has not yet been any association with stillbirth. Considering downregulated STB-specific genes in iSB, we also identified the *PSG* family, which are the most abundant placental proteins found in the maternal circulation during late pregnancy. The clinical relevance of measuring PSG levels in maternal serum was already suggested decades ago [45]. Early studies found that using anti-PSG antibodies or vaccination against PSGs induces abortion in mice and monkeys, and it reduced the fertility of non-pregnant monkeys [46,47]. Cui et al. analysed the differentially expressed proteins in the maternal serum in early recurrent spontaneous abortion, identifying PSG1 as a potential biomarker for recurrent pregnancy loss [48]. Another study analysing the Reactome pathway clustering showed that *PSG8*, *PSG1*, *PSG5*, *PSG4*, *PSG3*, *PSG7*, *PSG11*, and *PSG9* were involved in recurrent miscarriage [49]. However, until now, the expression of *PSG* has not been associated with stillbirth placentas. The expression of *PSG* genes is an early marker of human trophoblast differentiation into the syncytium development pathway. Camolotto et al. described a significant increase in total PSG proteins and transcripts in isolated CTB undergoing differentiation into the syncytium layer, supporting the obtained results [50]. Chen et al., using the choriocarcinoma BeWo cell line, identified another gene, *CRH*, capable of inducing trophoblast cell differentiation. Indeed, in this study, the authors suggested that *CRH* had no effect on apoptosis but promoted cell fusion and, consequently, syncytialisation [51]. We also noted a downregulation of *CRH* in the idiopathic cases analysed. Consequently, since these transcripts were reduced in the idiopathic cases, we hypothesised that the syncytiotrophoblast layer was apparently defective, potentially compromising placental function at the end of gestation. When we specifically examined DEGs resulting from iSB vs. LB, we identified more genes that supported our hypothesis. A minimum of eight genes (*ERVW1*, *ERVFRD-1* (syncytin-2), *ERVV-1*, *ERVV-2*, *ERVH48-1*, *ERVMER34-1*, *ERV3-1*, and *ERVK13-1*) encoding syncytin family members are known to be actively involved in trophoblast cell fusion and differentiation, playing a role in syncytialisation [52]. Curiously, *ERVFRD-1*, *ERVV-1*, *ERVV-2*, *ERV3-1*, and *ERVK13-1* were downregulated in iSB when compared with LB, as was the receptor for syncytin-2 (*MFSD2*) [53].

In a recent study that analysed the transcriptome profiles of placentas from spontaneous preterm births and pregnancy losses (including spontaneous miscarriages, recurrent miscarriages, and stillbirths), Wang et al. identified common molecules and pathogenic pathways involved in these compromised pregnancies [49]. When comparing stillbirths with full-term births, the top two enriched biological processes were DNA packaging and chromosome organisation, which were not observed in our GO analysis. On the other hand, *SOD2*, a key molecule involved in the cytokine pathway, was the only gene found to be upregulated in the different conditions studied (spontaneous preterm birth, stillbirth, and foetal facet from spontaneous miscarriage) [49]. Interestingly, when we compared iSB vs. LB, we also observed *SOD2* to be upregulated. Superoxide dismutase (SOD) is one of the main antioxidant enzymes, and its activity is considered to be a first line of defence against oxidative stress. The increased STB mitochondrial activity leads to the release of reactive oxygen species, consequently elevating oxidative stress, which induces the expression of antioxidants such as SOD [54]. The increased expression of *SOD2* can act as a compensatory response to increased oxidative stress in iSB compared to LB, suggesting that placental oxidative stress could be an aetiological factor involved in these cases. Since the best biomarkers of pre-eclampsia are associated with STB stress [55], it is plausible that unexplained foetal death could also encompass similar mechanisms. As placental dysfunction is typically diagnosed through histological examination of the placenta after delivery, it cannot be used as a marker before pregnancy loss. Many of the observed DEGs encode secreted proteins released into the maternal circulation; therefore, these transcripts could be regarded as candidates for plasma biomarkers of placental function to detect pregnancy problems and, consequently, prevent a stillbirth.

It is well known that CTBs produce low levels of hCG, while the STB expresses high levels of hCG [56], becoming readily detectable once a sufficient quantity of STB has developed [57]. Therefore, a failure of the STB layer in iSB would predict a lower secretion of hCG. Low hCG expression in placentas during the third trimester has been identified as an independent risk factor for intrauterine foetal demise after 34 weeks of gestation [58]. Our results showed that, in iSB, the observed signal of hCG was markedly lower, suggesting adverse effects on STB function or integrity. Interestingly, in iSB with FGR, hCG staining was preserved, but the number of syncytial knots observed appeared to be increased, which is consistent with placental insufficiency [59,60].

Maternal perception of reduced foetal movement (RFM) is associated with increased risk of stillbirth, and it has been reported that RFM represents a foetal compensation to conserve energy due to insufficient oxygen and nutrient transfer resulting from placental insufficiency. Lynne et al. observed that placentas from pregnancies with RFM had a higher density of syncytial knots and a greater proliferation index (Ki67), along with a decreased syncytiotrophoblast area, similar to what we observed in the iSB placentas analysed [60].

Placentas from pregnancies complicated by pre-eclampsia, FGR, or trisomy 21 have been associated with defective syncytialisation and altered expression of its modulators [61]. While much is known about histologically diagnosing placental insufficiency that can lead to stillbirth, the molecular aetiology remains unknown. The STB layer requires cellular modifications, including alterations in membrane fluidity, cell junctions, and cytoskeleton restructuring. Additionally, there is a biochemical differentiation that allows the STB to biosynthesise several proteins absent in the CTB. These proteins play distinct cellular functions, participating in metabolism, transport, and hormone production [61]. Since the results of our histological analysis were not as evident as the findings from the transcriptome analysis, our results collectively suggest that biochemical differentiation may be the primary defect occurring in unexplained stillbirths, affecting the metabolic and secretory activities of the STB. Indeed, the STB could play a fundamental role in maintaining pregnancy in the last trimester of gestation. Dysregulated STB formation disrupts the integrity of the placental exchange surface, reducing the ability of the foetus to compensate for subacute events in late pregnancy, which, in turn, can lead to FGR and intrauterine foetal demise.

In the future, overcoming challenges related to obtaining a sufficient number of high-quality samples with established clinical characteristics will be crucial for advancing the field and facilitating rigorous comparisons between different studies. There is an evident and urgent need for more research in the area of unexplained stillbirth. Understanding the reasons for pregnancy loss may aid parents in coping with their grief, assessing the likelihood of recurrent stillbirth, and identifying areas for preventive strategies in future pregnancies. The main limitation of our study was the number of samples that could be related to the observed dispersal of the PCA plot. It could have been foretold that, in fact, iSB (*n*= 4) would also be widespread, which was not the case. The wide dispersion of the SB samples could be due to the different aetiology of the SB samples, while all iSB were characterised with placental insufficiency. Nevertheless, our work reveals putative dysregulated genes that might underlie unexplained stillbirth, highlighting the importance of a detailed molecular characterisation to understand the mechanisms of reproductive success. This approach will ultimately provide new tools for the prevention, diagnosis, and interventions of pregnancy loss during the third trimester of gestation. The identified DEGs show promising results that may be valuable for the future discovery of biomarkers related to pregnancy losses. These signatures could also potentially serve as candidates for predictive markers of iSB, warranting testing in maternal blood throughout pregnancy. Indeed, studies have demonstrated the ability of plasma cell-free RNA to reveal patterns of normal pregnancy progression and determine the risk of developing pregnancy-related complications [62]. Furthermore, an open question remains as to whether each pregnancy complication is characterised by distinct aberrant placental gene expression profiles, or whether gestational disturbances also share common molecular signatures.

## 4. Materials and Methods

### 4.1. Ethics and Placental Sample Collection

This study was approved by the Ethics Committee of the Centro Hospitalar Universitário de São João (CHUSJ) and the Faculty of Medicine, University of Porto (FMUP) (Reference No. 430/19). Placental samples were collected at the Obstetrics Department of CHUSJ and processed at the Genetics Department of FMUP for diagnostic genetic analysis, with surplus material being stored for research purposes. Samples with chromosomal abnormalities, detected by karyotype or array comparative genomic hybridisation, were excluded from the study. Exclusion criteria included twin pregnancies, endocrine disorders, immunological diseases (such as antiphospholipid syndrome), inherited thrombophilia, pre-eclampsia, foetal cardiopathies, and pregnancies resulting from assisted reproductive technologies. All samples underwent evaluation by an anatomical pathology specialist and were divided into three groups: 3 term placentas from live births (LB), 5 placentas from stillbirths with an identified clinical cause (SB), including infections (*n* = 3) and umbilical cord pathologies (*n* = 2), and 4 placentas from idiopathic stillbirths (iSB), characterised by placental insufficiency with (*n* = 2) or without (*n* = 2) FGR. Detailed features of the samples are described in Table 1. In addition, placental samples were always collected from a region close to the insertion of the umbilical cord, so as to minimise sampling variation. Villous tissues were dissected under a stereomicroscope and washed with sterile PBS to remove excess blood. The chorionic villi were immersed in RNAlater (Invitrogen, Thermo Fisher Scientific Inc., Waltham, MA, USA) and were either immediately stored at −80 °C for RNA/DNA extraction or fixed in 10% formalin solution overnight for histochemical analysis.

### 4.2. RNA and DNA Extraction

To isolate RNA and DNA, the placental tissues were homogenised using Triple-Pure™ zirconium beads (Bertin Technologies, Montigny-le-Bretonneux, France) and a MiniLysis homogeniser (Bertin Technologies, Montigny-le-Bretonneux, France), with the addition of 1 mL of TRIzol reagent (Thermo Fisher Scientific Inc., Waltham, MA, USA). RNA and DNA isolation was performed following the manufacturer’s instructions. DNA purity and quantification were determined by a NanoDrop 2000 UV–vis Spectrophotometer (NanoDrop Technologies, Wilmington, DE, USA), and RNA quantification and integrity were measured by an Agilent 2100 Bioanalyzer (Agilent Technologies, Waldbronn, Germany). The RNA integrity numbers (RINs) ranged between 6.2 and 8.1, except for three samples that had values of 3.4 (SB), 4, and 5 (LBs). Additionally, exclusion of maternal contamination in the placental samples was performed using a quantitative fluorescent PCR using the Elucigene QST*R plus v2 kit (Elucigene Diagnostics, Manchester, UK), which analyses STR markers that allow the placenta DNA to be compared with corresponding foetal DNA.

### 4.3. RNA-Seq Library Preparation and Sequencing Analysis

A total of 12 placental tissue samples (*n* = 3 LB, *n* = 5 SB, and *n* = 4 iSB) were subjected to bulk transcriptomics. RNA-Seq libraries were prepared, and sequencing was performed by GenomeScan (https://www.genomescan.nl/; 1 September 2021). Briefly, poly-A selection was used for RNA-Seq on purified RNA (250 ng), the NovaSeq6000 (Illumina, San Diego, CA, USA) platform was used to sequence the libraries, and 150 bp paired-end reads were produced, delivered in two FASTQ files per sample. As UMI (unique molecular identifier) sequencing was performed, a third FASTQ file was produced for each sample. Raw FASTQ files were processed with UMI-tools [63] to incorporate the UMI information. Next, the files were processed with the BioWDL RNA-Seq pipeline v5.0.0 [64] developed at Leiden University Medical Center (LUMC). Briefly, the pipeline consisted of quality control (using FASTQC v0.11.9), adapter clipping (using CutAdapt v2.1), mapping reads to the human genome reference version GRCh38 (using STAR v2.7.3a), and expression quantification (using HTSeq-count v0.12.4). Counts were normalised using TMM and then log2-transformed (Appendix A). The heatmap was drawn using Complex Heatmap v2.6.2, and sample clustering was performed using the default Euclidean distance as a metric.

For downstream analysis, the gene expression count table was used as an input for an R (v4.1.0) script. In short, principal component analysis (PCA) was performed, and samples were visualised on a scatterplot using the first 2 principal components. Gene expression was visualised on heatmap plots. A differential gene expression analysis was performed (using DESeq2, v1.34.0), resulting in a table of differentially expressed genes (DEGs) (Appendix A). DEGs were considered to be statistically significant when the false discovery rate (FDR) value was below the threshold of 0.05. DEGs were visualised on a volcano plot generated based on -Log10(FDR) on the y-axis and avgLog2FoldChange on the x-axis. Furthermore, a functional enrichment analysis was performed on the DEG list (using gprofiler2, v0.2.1).

Additionally, total RNA was subjected to treatment with DNase I (Thermo Scientific Inc., Waltham, MA, USA) and cDNA was synthesised using 1 µg of the DNase-treated total RNA and 4 µL of qScriptTM cDNA SuperMix (Quanta Biosciences, Inc., Beverly, MA, USA). The master mix used was the 2 × PowerUp SYBR Green Master Mix (Thermo Scientific Inc., Waltham, MA, USA), following the manufacturer’s instructions (Appendix A).

### 4.4. Pathway Enrichment Analysis

Gene Ontology (GO) term enrichment was analysed using the gprofiler2 package [65]. GO analysis allowed us to explore the gene functions of DEGs based on different annotated categories: biological processes (BP) and enriched pathways by Kyoto Encyclopedia of Genes and Genomes (KEGG) analysis. Additionally, the molecular functions (MF), cellular components (CC), and Reactome (REAC) analysis can be found in the Appendix A. GO terms with *p* < 0.05 were considered significant.

### 4.5. Cell-Type-Specific Gene Expression Signatures

To assess whether iSB’s downregulated DEGs were specifically expressed in the STB, we used processed RNA-Seq data from isolated primary trophoblast cell types [29], comparing normalised gene expression values between the CTB and STB. To obtain a more comprehensive view of how gene expression changes were related to placental cell composition, we used processed single-cell RNA-Seq data from term placentas [30]. We extracted marker genes for each cell type and plotted their expression fold-change (log2) between iSB and control samples. A deviation from zero (no change) was assessed using one-sample *t*-tests, followed by correction for multiple comparisons using the Benjamini–Hochberg method.

### 4.6. Immunohistochemistry and Immunofluorescence

For immunohistochemical analysis, sections from formalin-fixed paraffin-embedded tissue blocks were deparaffinised and hydrated, using standard procedures. Primary antibodies for hCG (rabbit polyclonal, A0231, diluted 1:500, DAKO, Glostrup, Denmark) and Ki-67 (rabbit monoclonal, 30-9, Roche, Basel, Switzerland) were used in combination with the OptiView DAB IHQ detection kit according to the manufacturer’s instructions on a VENTANA benchMark ULTRA platform (Ventana Medical Systems, Tucson, AZ, USA). Adequate positive controls were included.

For immunofluorescence, sections were deparaffinised, and antigen retrieval was carried out with Tris–EDTA buffer (10 mM Tris, 1 mM EDTA solution, pH 9.0) for 12 min at 98 °C in a microwave. After antigen retrieval, the sections were left to cool down, washed twice for 5 min with PBS and once with 0.05% Tween-20 (822184, Merck, Darmstadt, Germany) in PBS (PBST) for 5 min, and treated for 1 h in blocking buffer [1% bovine serum albumin (BSA) (A8022-100G, Life Technologies, Carlsbad, CA, USA) in PBST] at room temperature. For the rest of the procedure, the slides were in a humidified chamber. After blocking, the sections were incubated with primary antibodies, diluted in blocking buffer overnight at 4 °C, washed twice with PBS, once in PBST, and incubated with secondary antibodies diluted in blocking buffer containing 4′,6-diamidino-2-phenyl-indole (DAPI) (1:1000, D1306, Life Technologies, Carlsbad, CA, USA) for 1 h at room temperature. The primary antibody combinations used were sheep anti-COL1 (1:200, AF6220, R&D system, Minneapolis, MN, USA), mouse anti-HLA-G (1:100, 11-291-C100, Exbio, Vestec, Czech Republic), and rabbit anti-Ki67 (1:100, ab15580, Abcam, Cambridge, UK). The secondary antibodies used were Alexa Fluor 488 donkey anti-mouse IgG (1:500, A21202, Life Technologies, Carlsbad, CA, USA), Alexa Fluor 594 donkey anti-rabbit IgG (1:500, A21207, Life Technologies, Carlsbad, CA, USA), and Alexa Fluor 647 donkey anti-sheep IgG (1:500, A21448, Life Technologies, Carlsbad, CA, USA). The negative control for Ki67, HLA-G, and COL1 was obtained by omitting the primary antibodies (Appendix A). Finally, the slides were washed and mounted with Pro-Long Gold (P36930, Life Technologies, Carlsbad, CA, USA).

### 4.7. Imaging Analysis

Whole-slide images (WSIs) from immunohistochemistry slides were obtained using a Hamamatsu S360 MDEU scanner (40-fold magnification) (Hamamatsu Photonics, Hamamatsu City, Japan), and uncompressed native WSI files were analysed with the NDP.view2 software. For immunohistochemistry slides, an average of 10 fields of view was randomly chosen for image evaluation of each placental sample, using a light microscope. The section chosen for capture was in a comparable area on the immunohistochemistry and immunofluorescence slides. Images from immunofluorescence slides were captured and assessed using the ZEISS Axio Scan.Z1 Slide Scanner (ZEISS, Oberkochen, Germany).

### 4.8. Statistical Analysis

The two-tailed Student’s *t*-test by IBM SPSS Statistics (v29.0) was used to statistically compare groups for each variable (gestational age and maternal age). Data are given as the mean ± standard deviation. Differences were considered to be statistically significant at values of *p* < 0.05. The PCA plot and hierarchical clustering (Euclidian correlation as the distance function) were performed in R (R version 4.1.0). Differential gene expression in RNA-Seq data was tested using DESeq2 (v1.34.0), and the raw *p*-values obtained were adjusted using the Benjamini–Hochberg procedure to control the FDR. A gene was considered to be differentially expressed when the FDR value was below the threshold of 0.05. Additionally, GO terms with *p* < 0.05 were also considered significant. Finally, the *t*-tests, corrected for multiple comparisons using the Benjamini–Hochberg method, were used to compare our dataset with other RNA-Seq datasets available online.

## Figures and Tables

**Figure 1 ijms-25-05180-f001:**
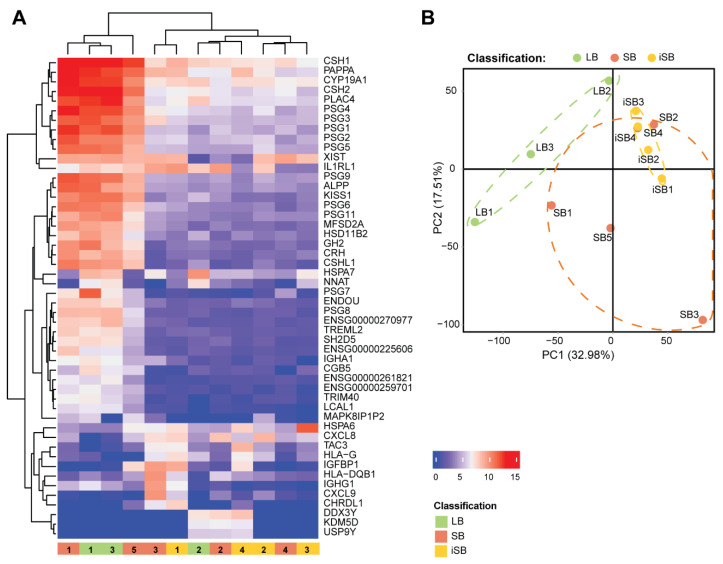
Hierarchical clustering analysis of gene expression data of placental samples. Heatmap of the top 50 most highly variable genes across all placental samples (**A**), and PCA plot for clustering gene expression data according to biological groups (**B**). Red represents genes with high expression levels, while blue represents genes with low expression levels. iSB—idiopathic stillbirth; SB—stillbirth (with a known cause); LB—live birth.

**Figure 2 ijms-25-05180-f002:**
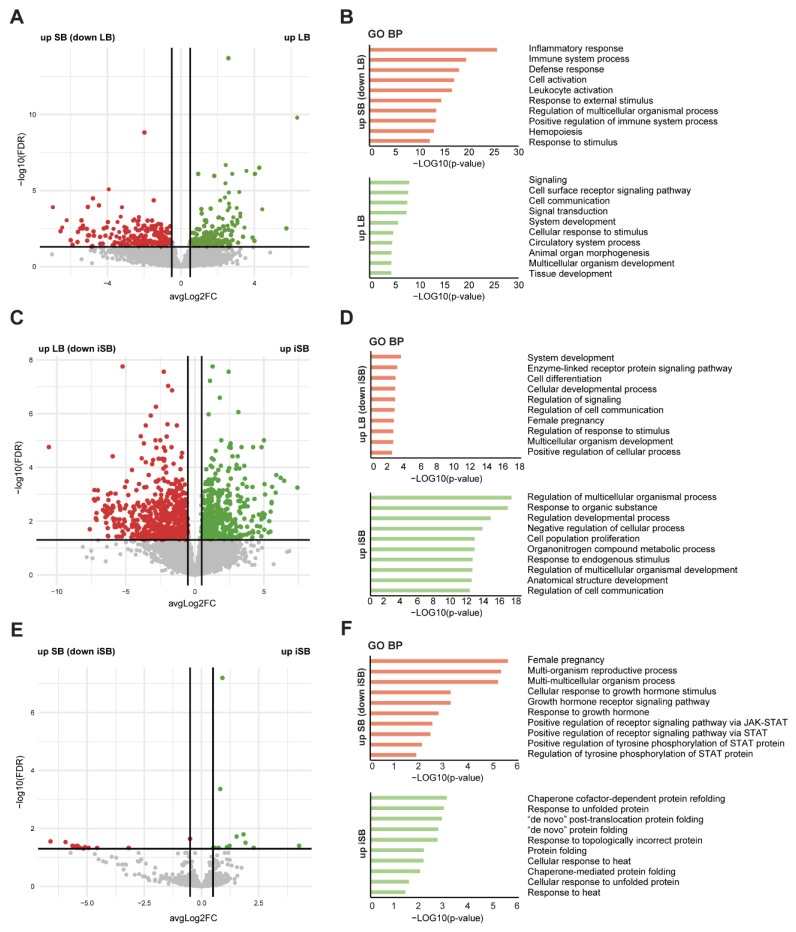
Differential gene expression analysis: Volcano plots of gene expression profile data in SB vs. LB (**A**), iSB vs. LB (**C**), and iSB vs. SB (**E**). The thick horizontal line represents the threshold of FDR < 0.05. Colour coding is based on this threshold and represents the differentially expressed genes (DEGs) between the respective biological groups. Red dots indicate significantly under-expressed genes, while green dots indicate significantly over-expressed genes. Gene Ontology (GO) enrichment analysis of biological processes (BP) of downregulated (red bars) and upregulated (green bars) DEGs for the three comparisons studied: SB vs. LB (**B**), iSB vs. LB (**D**), and iSB vs. SB (**F**). Top 10 of GO for BP categories sorted by decreasing order of *p*-values, based on the GO enrichment test *p*-value. iSB—idiopathic stillbirth; SB—stillbirth (with a known cause); LB—live birth.

**Figure 3 ijms-25-05180-f003:**
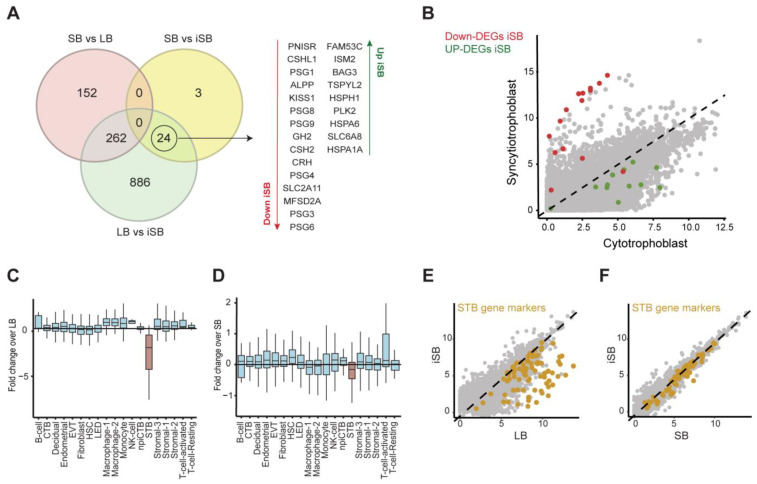
Idiopathic stillbirth placental samples showed defects in STB markers. Venn diagram showing the overlap of differentially expressed genes (DEGs) among the three comparisons studied: SB vs. LB, SB vs. iSB, and LB vs. iSB. On the right is a list of deregulated genes in iSB samples when compared with LB or SB (**A**). The expression of downregulated genes (red) and upregulated genes (green) in the STB and CTB in iSB are represented in a scatterplot (**B**). Boxplots summarising the estimated cell type proportions of term placentas for each of the 19 cell types, which are encoded on the x-axis. The fold-change values for cell type proportions are encoded on the y-axis. The same training dataset was used for each comparison: LB vs. iSB (**C**) and SB vs. iSB (**D**). Scatterplots showing the plotting of marker genes’ expression in the STB for both analyses: LB vs. iSB (**E**) and SB vs. iSB (**F**). CTB: cytotrophoblasts; EVT: extravillous trophoblasts; LED: lymphoid endothelial decidual cell; HSC: hematopoietic stem cell; npiCTB: non-proliferative interstitial cytotrophoblast; STB: syncytiotrophoblast; iSB—idiopathic stillbirth; SB—stillbirth (with a known cause); LB—live birth.

**Figure 4 ijms-25-05180-f004:**
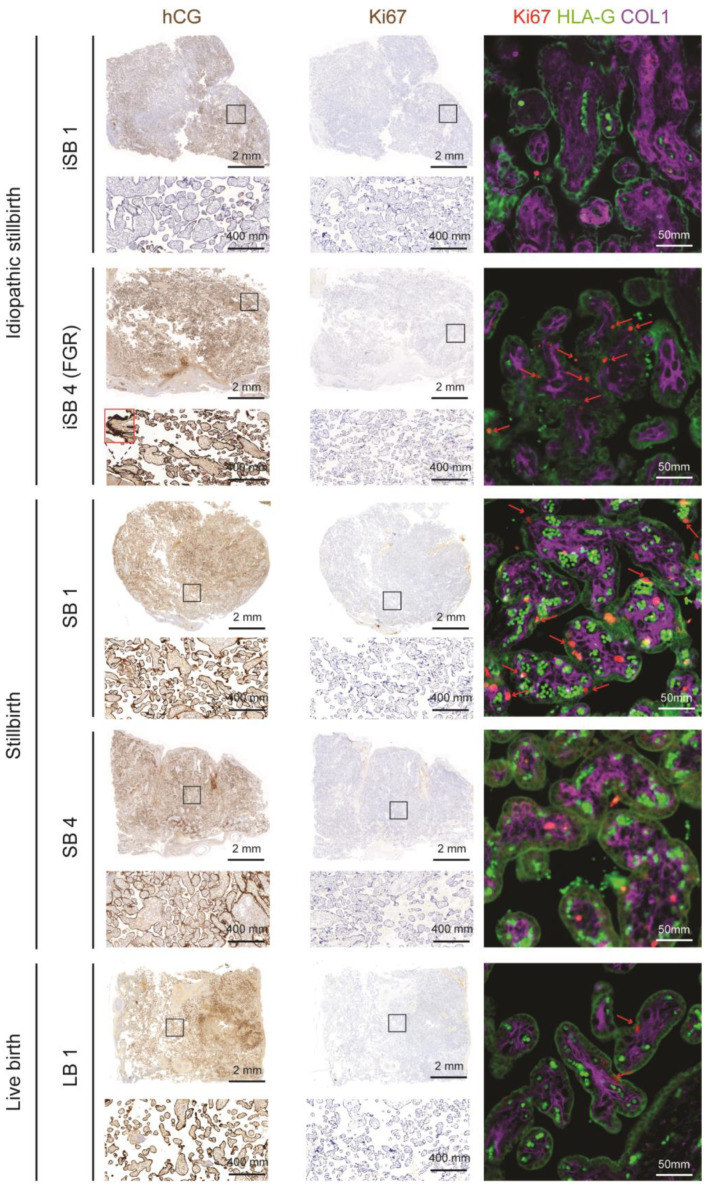
Immunohistochemistry and immunofluorescence staining on placentas from stillbirths and live births for proteins associated with CTB proliferation and STB, namely, Ki67 and hCG, respectively. Immunohistochemical staining for hCG and Ki67 in placental tissue sections, where the positive staining of hCG and Ki67 is brown. An immunofluorescence staining combination is also shown—Ki67 (red) + HLA-G (green) + Col1 (purple). The black squares denote the field of tissue that has been captured, the zoomed red square is an example of syncytial knot, and red arrows point to positive Ki67 staining on immunofluorescence. The scale bars are represented on the right inferior side of each image.

**Table 1 ijms-25-05180-t001:** Sample characteristics: Clinical features and medical history of placental samples.

Sample ID	Maternal Age (Years)	Fetal Sex	Karyotype/aCGH Results	Gestational Age (Weeks)	Previous Obstetric History	Maternal Characteristics	Clinical Findings
**LB 1**	28	F	NA	38	3G1P1A	Non-smoker and maternal obesity.	Natural childbirth
**LB 2**	28	M	NA	36 + 6 days	1G	Non-smoker	Natural childbirth
**LB 3**	38	F	NA	37	2G1P	Smoker (4 cigarettes/day)	Placenta previa, cesarean section
**SB 1**	31	F	46,XX	34	2G1P	NA	Infection cause
**SB 2**	28	M	arr(X,Y)x1,(1-22)x2	39	3G2P	NA	Infection cause
**SB 3**	35	F	46,XX	26	3G2P	NA	Lympho-histiocytic villitis of unclear etiology, very likely of infectious origin
**SB 4**	32	F	46,XX	31	3G1P1A	Smoker, maternal obesity and endometriosis	Umbilical cord torsion
**SB 5**	25	F	46,XX	35 + 6 days	1G	NA	True Knots in umbilical cord
**iSB 1**	39	F	46,XX	35	2G1A	Hypertension	Placental insufficiency
**iSB 2**	24	F	46,XX	36	2G1A	NA	Placental insufficiency
**iSB 3**	33	F	46,XX	28	2G1A	Smoker (8 cigarettes/day)	Placental insufficiency and FGR
**iSB 4**	39	M	arr(X,Y)x1,(1-22)x2	30	3G1P1A	Maternal obesity	Placental insufficiency and FGR

LB—live birth, SB—stillbirth, iSB—idiopathic stillbirth, F—female, M—male, NA—not available, G—gravida, P—para (parity); A—abortion; FGR—foetal growth restriction.

## Data Availability

Data are contained within the article and Appendix A.

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
