# Peer review of "Syncytiotrophoblast Markers Are Downregulated in Placentas from Idiopathic Stillbirths"

_ijms, 2024, doi:10.3390/ijms25105180_

Round 1

Reviewer 1 Report

Comments and Suggestions for Authors

The paper “Syncytiotrophoblast markers are downregulated in placentas from idiopathic stillbirths” provides novel and interesting information on the potential idiopathic stillbirth marker candidates. The Authors used the NGS method as well as bioinformatic tools that allowed them not only to present the data from the sequencing of placental samples but also to compare them with previously single-cell sequencing results to assign markers to the specific cell types. Furthermore, the Authors performed the immunostainings of placental samples which at least partly confirm the NGS results. I found the paper of good quality but I wish the Authors to make some corrections which in my opinion will be beneficial. Please, find the list of my concerns below.

Major concerns:

1.        Please explain each shortcut the first time used. Some gene names are used in the results section but are explained in the Material and Methods section.

2.        I would like the Authors to compare and describe GO terms revealed to be affected in the iSB group in the main text, not only in the figure. You should also add the information on examples of DEGs annotated to those GOs. Further, you showed only GO(BP). What about MF and CC? I found no information on affected KEGG pathways. Last but not least, you mentioned Reactome analysis but provided no results from this analysis. I think these results should be described and discussed.

3.        In lines 248 and 250 as well as in others you mention specific GO terms. Please provide examples of DEGs annotated to these terms.

4.        Please, write all gene symbols in italics to distinguish them from proteins.

5.        Line 333: “intensity”: is not the right word here, please, rephrase. Anyway, have you compared hCG blood levels in women after LB with SB and iSB?

6.        Lines 339-341: IHC/F-IHC is not a quantitative method and may not be a basis for protein content evaluation. Please rephrase.

7.        Lines 425-426: Samples with RIN value under 8 are of doubtful quality. Especially if your experiment is based on controls n=3 and two of these samples are of critically low quality. This raises the question if the controls are appropriate. In my opinion, you should provide the results of the validation study with the use of qPCR, at least for the genes you widely discuss in the study.

8.        There are no proper negative controls of IHC/F-IHC provided. Please, provide the pictures of appropriately performed controls: with the use of blocking peptide, incubation of primary antibody with an excess of antigens before staining or at least omitting primary antibodies.

Minor issues:

1.        Lines 25-28: Divide this sentence into 2-3 shorter and rephrase it as it is difficult to understand

2.        Line 137: “of” not “on”

3.        In the main text I found lots of double spaces. Please correct it.

4.        Line 415: what do you mean by “Para”? Please, explain.

5.        Material and methods: please, provide a country of each supplier origin and unify the information on them across the section.

6.        Line 444: “was” not “as”

Comments on the Quality of English Language

only a few editorial errors.

Reviewer 2 Report

Comments and Suggestions for Authors

In this work, authors examined and compared placentas from live births (LB), still births (SB) and idiopathic stillbirths (iSB), and concluded that idiopathic stillbirths’ placentas show upregulated and/or downregulated different genes and markers, indicating anomalies of syncytiotrophoblast in iSB placentas.

-This is a beautiful work, with beautiful images. Interesting topic. There are some MAJOR REVISIONS the authors could do, in order to improve the paper:

-please write the reference numbers in square brackets in the text, like this:[7], instead of “(Smith et al, year)”

-in lines 118-120 you wrote:” Indeed, principal component analysis  (PCA) showed considerable heterogeneity in SB and LB placental samples, however our study iSB cases were located mainly in the same quadrant (Figure 1B).” Actually, in Figure 1B, the study iSB cases were in two quadrants, and so were the LB cases.  Please rephrase.

-in lines 178-181 you wrote:” This showed that iSB down- DEGs were specifically expressed in STB, whereas iSB up-DEGs were preferentially expressed in CTB, as would be expected if there was an underrepresentation of STB in iSB  placentas”. How did you come to this conclusion:” as would be expected if there was an underrepresentation of STB in iSB  placentas”? Do you have a reference for this?

-in lines 209-210 you wrote:” Ten fields of view were chosen at random to evaluate the images of each placenta sample…”Ok. From the TEN fields of view from each placenta, how did you conclude/choose the ONE image for each placenta in figure 4? The most colored/uncolored ones by the staining that you were following? Because, for example, for Ki67, in iSB 1 there is ONE red dot, and in LB 1 there are TWO. If you change the field of view near the image in LB 1, you might find a field of view with only ONE red dot, like iSB. From my point of view, there is no clear difference between these two images.

-in line 212 you wrote:” In placentas from SB and from LB, hCG staining was normally found on the membranes of the STB layer of the villus surface.” Actually, the red color of hCG staining was normal in LB 1 and SB 4, while in SB 1 there is similar image as in iSB 1. You described in lines 215-216:” Regarding the other iSB  case, we observed focal areas without hCG staining interspersed with areas that stained weakly. “ True. But this description is true for SB 1 , too. Please check and correct.

-in figure 4 there is a huge difference in the hCG staining of iSB (with FGR) and iSB 1. Maybe you could write a possible explanation about this.

-in lines 217-221 you wrote:” The term enrichment ‘cell population proliferation’ in iSB also led us to evaluate the proliferative marker Ki67 staining, which marked proliferative CTB. Indeed, it is expected to observe a residual number of cytotrophoblasts in a term placenta, as seen in LB. However, we observed more positive CTB in iSB with FGR comparing to LB but an absolute quantification would be necessary to confirm this increase.” Actually, when looking at the red dots of Ki67 staining, the striking perception is that iSB 1 is quite similar to LB (one red dot versus two red dots), while the iSB (with FGR) shows A LOT of red dots; SB 1 and SB 4 seem to have a moderate number of red dots of Ki67 staining. So that you above conclusion; “Indeed, it is expected to observe a residual number of cytotrophoblasts in a term placenta, as seen in LB” could be completed with the astonishing same number in iSB 1, totally unexpected and unexplained.

-in lines 221-222 you concluded: “Together, the results suggest that there’s less viable STB in iSB, however it is affected of different pathways…”Together whom to whom? The red dots of Ki67 do not suggest that. The gene expression analysis does suggest that.

-in lines 420-421 you wrote:” RNA and DNA isolation were performed following manufacturer's instructions, with the exception of DNA precipitation at −20°C that was 2 hours after the addition of absolute ethanol. DNA”. What do you mean by “with the exception of”? That means that you did not follow entirely the instructions, and the results are not valid. Please rephrase.

-in lines 502-507 you wrote:” Whole slide images (WSI) from immunohistochemistry slides were obtained using a Hamamatsu S360 MDEU scanner (40-fold magnification) and uncompressed native WSI files were analysed with the NDP.view2 software. For immunohistochemistry slides, an  average of 10 fields of view was randomly chosen for image evaluation of each placental sample, using a light microscope. The chosen section to capture was in a comparable area on immunohistochemistry and immunofluorescence slides. Images from immunofluorescence slides were captured and assessed using ZEISS Axio Scan.Z1 Slide Scanner.” Ok. But how did you COMPARE the microscope images? To say that the Ki67 for example is overexpressed (many red dots) in one placenta versus another/ in one field versus another? Was there an author, with high experience, to compare it? Was it a software? It is not clear. Please explain.

-in References, there are only 21 articles from 2019 and more recent, out of 65 titles: 6,7,9,14,18,19,20,21,24,25,31,38,40,43,44,49,56,59,60,65. Please replace most the older ones with some more recent titles.

Round 2

Reviewer 1 Report

Comments and Suggestions for Authors

The Authors addressed all my comments. I have no other issues.

Reviewer 2 Report

Comments and Suggestions for Authors

Thank you for the changes you made. It is ok now.